# Influence of Y Nano-Oxide and Its Secondary Phase on Microstructure, Mechanical Properties, and Wear Behavior of the Stainless Steel Coatings Fabricated by Plasma Transfer Arc

**Junyu Yue [1], Yi Sui [1,\*], Lifeng Yang [2], Fei Lu [1], Weidong Chen [3,\*], Xiaoyu Liu [1] and Xiaohua Sun [1]**

1   State Key Laboratory of Baiyunobo Rare Earth Resource Researches and Comprehensive Utilization, Baotou Research Institute of Rare Earths, Baotou 014030, China; btyuejunyu1994@163.com (J.Y.); yurifeiphy@126.com (F.L.); xiaoyu_liu1976@163.com (X.L.); btsuiyi@163.com (X.S.)
2   Inner Mongolia Autonomous Region Key Laboratory for Testing and Measure Technology of Special Steel and Products, Baotou 014033, China; yanglifengwo@163.com
3   School of Materials Science and Engineering, Inner Mongolia University of Technology, Hohhot 010051, China
\*   Correspondence: btsuiyi@126.com (Y.S.); weidongch@163.com (W.C.)

**Abstract:** Rare-earth is an efficient refiner for surface modification of steel material. This study presents the synergistic influence of $Y_2O_3$ nanoparticles (YNPs) and Mn-oxide secondary phase on the microstructure and mechanical properties of 14CrSiMnV coating fabricated by plasma transfer arc cladding process. The results indicated that the residual Y accumulated with Mn, forming a secondary phase oxide particle instead of inclusions or slags during rapid cooling solidification of the coating. Due to enlarged equiaxed grains, declining long-range dendritic grains, and secondary phase strengthening, steel coatings present hybrid-type fracture mechanism, less plastic deformation, and third-body interaction. With an optimal addition of YNPs (0.4 wt.%), the mechanical properties of the steel coating are improved, as indicated by the increase of 92.0% in the tensile strength, increase of 55.6% in the elongation, increase of 11.3% in the microhardness, decrease of 22.2% in wear weight loss, and increase of 28.3% in relative wear resistance.

**Keywords:** plasma transfer arc; steel coating; $Y_2O_3$ nanoparticles; Mn secondary phase

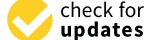

## 1. Introduction

In recent years, a great deal of study attention has been paid to rare-earth (RE) materials, such as La [1,2], Ce [3,4], or Y [5,6], owing to their applications in additive manufacturing and surface engineering for declining the formation of dendritic solidification microstructure, non-equilibrium segregations, as well as pores and defects of alloy material during the rapid solidification process. As a result, brittle fracture, cracks, and wear debris under loading or dry friction conditions are decreased, whereas the wear resistance, hardness, strength, and toughness of the steel material are improved with small amounts of RE material [7–9].

However, due to the strong affinity with O and S, the excessive addition of RE leads to enrichment of RE at grain boundaries to form inclusions, compromising the overall performance of materials [10–12]. One the other hand, progressive fabrication technologies and sufficient material performances of steel material are required to meet the rapid development and complex service conditions in cylinder heads, automotive, and train braking systems [13–16]. Therefore, an optimal addition strategy of RE is desired for high stability and cost-effectiveness application in the field of refinement for commercial surface modification, such as laser direct-deposited (LDD) and plasma-transferred arc (PTA) technology.

It is reported that secondary phase act as preferential crack nucleation sites [17] and Zener-pinning particles [18] in steel materials, providing a lower crack-free path to drop the

absorbed energy and a dragging force to retard grain growth, thereby playing an effective role in crack-free path and grain refinement, and resulting in better tensile properties and strength. In this way, RE-based secondary phases decline the undercooling for heterogenous nucleation and lead to significant grain refinement and microstructural homogeneity of alloy [1,19], drawing an increasing attention in rapid solidification process. Meanwhile, some studies have suggested that added RE material could form secondary phase oxide with alloyed element (such as Si and Ti) instead of inclusions or slags [20–24], which possess a positive influence on grain refinement, grain boundary improvement, structural transformation, and microalloying, inhibiting the growth of columnar crystals and crack reduction in alloy.

A small amount of Mn as beneficial element which leads to grain refinement, precipitation hardening [25], and stacking fault [26], improving the eutectic morphology, strength and strain of the steel materials. Furthermore, Mn is a potential candidate to form secondary phase particles for the accumulation of Y during the fast solidification process, while the Y (atomic radii is 0.180 nm, ionic radii is 0.09 nm) and Mn (atomic radii is 0.127 nm, ionic radii is 0.058–0.0645 nm) are suitable to form a substitutional solid solution with Fe (atomic radii is 0.172 nm, ionic radii is 0.055–0.0645 nm). However, the combined effect of Y and Mn elements within high-strength steel has not been studied in detail.

The purpose of the work is to find the optimal surface modification method of PTA to add $Y_2O_3$ material and alloying of the cast iron coating to improve its performance properties. The synergistic influence of a small amounts of Y and Mn-based secondary phase on the microstructure and mechanical performance were investigated. Moreover, $Y_2O_3$ nanoparticles (YNPs) were adopted to improve the dispersity of Y element. Meanwhile, the optimal content of YNP addition is determined by microhardness, tensile, and wear performance.

## 2. Materials and Methods

### 2.1. Materials

As shown in Figure 1a,b, the particle size of 14CrSiMnV steel powder (GL PTA Inc., Wuhan, China) is 80–120 μm. The composition of steel powder was 14.5 wt.% Cr, 1.2 wt.% Si, 0.5 wt.% Mn, 0.5 wt.% V, 0.3 wt.% C, 0.05 wt.% O, and Fe (bal.). Before the PTA process, the powder was dried to remove the moisture by heating at 80 °C.

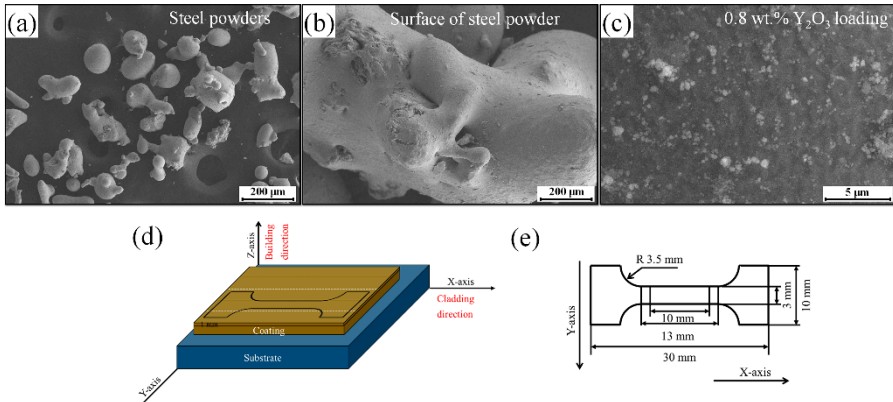

**Figure 1.** (**a**) Macro view of 14CrSiMnV steel powder, (**b**) Morphology of 14CrSiMnV powder, (**c**) Surface of 14CrSiMnV powder covered with $Y_2O_3$ nanoparticles, (**d**) Location of tensile specimens extracted from the coatings, (**e**) schematic illustration of tensile specimens.

YNPs precursor was prepared by applying the reverse precipitation method within 2 h from $Y(NO_3)_3$ (analytic reagent) and $NH_4HCO_3$ (analytic reagent). Consequently, a washing step and a freeze-drying process were enforced to obtain the low-impurity and dispersed powder. To obtain yttrium oxide, the precursor was heated at the heating rate of

5 °C/min and kept at 500 °C in the air atmosphere. The diameter of YNPs ranged from 30 to 50 nm (Figure 1c).

YNPs and 14CrSiMnV steel powder were mixed by a planetary grinding machine. After mixing at 200 rpm for 40 min, the surface of steel powder was coated with different amounts of YNPs (0 wt.%, 0.2 wt.%, 0.4 wt.%, 0.6 wt.%, and 0.8 wt.%), as shown in Figure 1c. Then, the mixed powder was cladded into high-strength steel coating (abbreviated as HSC), which are labeled as HSC0, HSC02, HSC04, HSC06, and HSC08, respectively.

## 2.2. PTA Coating Preparation

Q235 steel plate ($200 \times 100 \times 10$ mm$^3$) was adopted as a substrate. The plates were ground using 180 grinding wheels and degreased with anhydrous ethanol. The composition of the Q235 was $\leq$0.17 wt.% C, $\leq$0.35 wt.% Si, $\leq$0.14 wt.% Mn, $\leq$0.035 wt.% P, $\leq$0.035 wt.% S, and Fe (bal.). The coating sample ($50 \times 55 \times 4$ mm$^3$) was manufactured by 7 channels of the single-channel cladding layer ($50 \times 12 \times 4$ mm$^3$) with a plasma-transferred arc machine (PTA, PTA-BX-400b), as shown in Figure 1d. The 14CrSiMnV coating is metallurgically bonded to the substrate. To achieve a relatively smooth surface of the coating and less significant substrate deformation, the processing parameters are conducted as shown in Table 1. Then, the coating specimens were cut from PTA coatings and ground using SiC abrasive paper (180, 320, 500, and 800 grit). The metallographic samples were polished with a diamond polishing paste (1 μm), followed by etching with an amygdalic acid (1 g/100 mL) and hydrochloric acid (5 mL/100 mL) ethanol solution.

**Table 1.** Experimental process parameters.

| EL | Parameter |
| --- | --- |
| Arc length (mm) | 10 |
| Current (A) | 170 |
| Plasma gas | Argon |
| Plasma gas flow (L/min) | 5 |
| Protective gas flow (L/min) | 8 |
| Feeding gas flow (L/min) | 5 |
| Powder feeding rate (mg/s) | 400 |
| Scanning velocity (mm/s) | 1 |
| Overlap rate (%) | 40 |

## 2.3. Material Characterization

The surface morphology and microstructure of the powders and coatings were characterized with field-emission scanning electron microscopy (FE-SEM, Zeiss-Sigma500) and optical microscopy (OM, Zeiss Imager-A1m). In addition, the local phase structure was identified with X-ray diffraction (XRD), using Cu Kα radiations (λ = 0.15406 nm) with a scan rate of 5°/min in the 2θ range of 20° to 80°. Moreover, the oxygen and nitrogen determinate (OND, HORIBA-EMGA820) was adopted to measure the oxygen content. The method is high frequency heating combustion–infrared absorption. The area size for analysis is 0.010~0.5%. The power is 8 kW and the temperature is 3500 °C. ICP mass spectrometer (PerkinElmer-Nexion300Q) was used to measure the yttrium content. The method is inductively coupled plasma mass spectrometry. The area size for analysis is 0.0005~0.10%.

## 2.4. Mechanical Characterization

The microhardness of coating was recorded with a Vicker's hardness tester (Everone EM-4500 digital). To analyze the average microhardness value, three independent points were recorded under a load of 100 g for 15 s, which was taken along the cross section of coating.

Tensile properties of the coating were determined with a universal testing machine (AG-Xplus, Shimadzu, guangzhou, china). The test sample was cut along the *x*-axis, using

a non-proportional sample style of GB 228 (Chinese standard), as shown in Figure 1e. The tensile sheet sample was selected on the upper surface of the 14CrSiMnV alloy coating with a thickness of 1 mm ($\pm$0.05 mm). The sample was loaded with extension rate of 0.5 mm/min.

The reciprocating wear process is a reasonable simulation pattern to simulate the actual working condition, while the sample needs to work continuously and stably for a long time. A surface performance tester (MFT-4000, Lanzhou, china) was employed to test the reciprocating friction of 14Cr2NiSiVMn coating. The surface performance tester and the schematic diagram of the experimental sample are shown in Figure 2a. Coating samples were polished before the experiment to avoid the influence of coating surface roughness on experimental results. To obtain average value as a repeatability test result, each sample was tested with three groups of wear. To avoid introducing new elements and affecting wear results in this experiment, Si3N4 ceramic ball ($\Phi$ 3 mm, 2200 Hv) was employed as the friction pairs. The reciprocating friction test parameters were set as Table 2. Parallel tracks relative to the cladding direction were carried out for each specimen, as shown in Figure 2b.

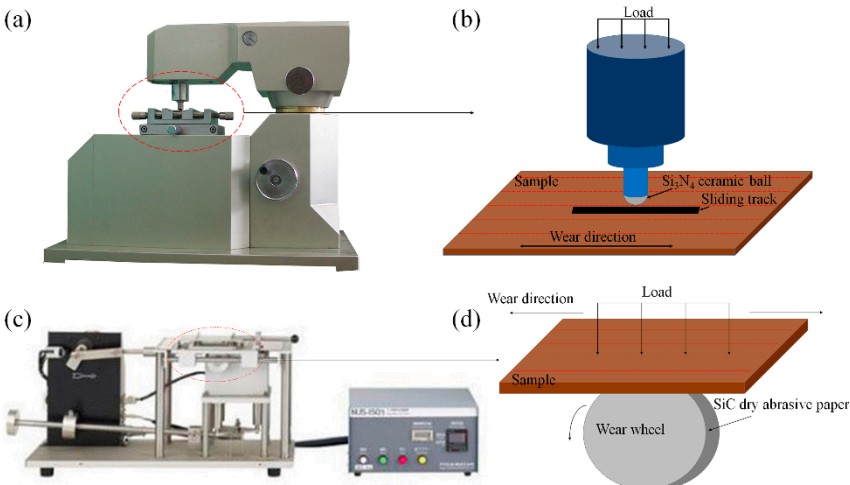

**Figure 2.** (**a**) The surface performance tester MFT-4000, (**b**) the schematic diagram of the experimental sample, (**c**) the wear tester NUS-ISO3, (**d**) the schematic diagram of the experimental sample.

**Table 2.** The wear test parameters.

| EL | The Reciprocating Friction Test | The Wear Weight Loss Test |
|---|---|---|
| Wear load (N) | 10 | 18 |
| Wear speed (mm/min) | 50 | 60 |
| Length (mm) | 5 | 30 $\times$ 12 |
| Temperature (°C) | 25 | 25 |
| Relative humidity (%) | 17 | 17 |
| Friction state | Dry friction | Dry friction |
| Contact form | Point-surface | Line-surface |

Wear weight loss is the most important test method for wear-resistant alloy materials. To more intuitively show the influence of CNRs content on wear resistance of 14Cr2NiSiVMn alloy coating, the weight loss test of PTA alloy coating was carried out by using a wear tester (NUS–ISO3, Hapoin, Shanghai, China). The wear tester and schematic diagram of the experimental sample are shown in Figure 2c,d. To determine the weight-loss change of samples, the weight of samples before and after wear were measured. In this way, five wear points were recorded, then the difference was calculated. To obtain average

value as a repeatability test result, each sample was tested with three groups of wear. The specific formula is as follows:

$$\text{Weight loss} = W_{\text{before wear}} - W_{\text{after wear}} \tag{1}$$

A specific standard sample is usually selected as a reference for the characterization of the relative wear resistance ($\varepsilon$). The specific formula is as follows:

$$\varepsilon = W_{\text{sample}} / W_{\text{standard}} \tag{2}$$

In this paper, W represents the wear weight loss. The samples are PTA alloy coating, and the standard sample is Q235 steel.

Influence of errors was reduced by fine polishing of coating sample with 2500 mesh sandpaper, surface clean with alcohol ultrasonic waves and surface dry with a blow-dryer before weighing. An electronic balance (BSA224S, accuracy of 0.0001 g) was used for weighing. To avoid the introduction of new elements, which would affect the wear results, 80 mesh SiC dry abrasive paper (2265.3 $\pm$ 224.5 Hv) was employed as the friction pairs. The wear test parameters were set as Table 2, which conforms to Japanese Industrial Standards (No. JIS H8503-1989). The wear wheel will carry out a new friction surface on the lower side of the test sample for each scratch path by rotated 0.9° in the next scratch, preventing the influence of SiC sandpaper surface on the test results.

## 3. Results and Discussion

### 3.1. Subsection Microstructural Characterization

Figure 3 presents the metallographic morphology along the cross-section of HSC specimens. In the unmodified coating, the long-range dendritic solidification microstructure growing along certain angles can be clearly observed, as well as several larger defect points, as shown in Figure 3a,b [27]. However, long-term dendritic solidifications are reduced and equiaxed grains are enlarged in the specimen modified with 0.4 wt.% YNPs, while the concentration of internal defects is significantly reduced, as shown in Figure 3c,d. The RE element can improve the undercooling point during the solidification of alloy system, thereby reducing the metastable eutectic transition temperature [22,28]. During the rapid heating process, dispersed Y-ions partially dissolve from part of YNPs with low solubility, segregating at the front of primary dendrites with low electronegativity, reducing the undercooling degree of the steel system, and providing an enhanced nucleation rate [29]. In addition, YNPs improve the fluidity of melt in the molten pool, resulting in multi-directional heat flow at all locations in the molten pool. The dendrite phase is reduced as dendrites dissociated into smaller species by undercooling effect, resulting in grain refinement and multi-directional crystallization. Consequently, long-term dendritic solidifications are transformed into equiaxed grains and short-term columnar crystals, resulting in a dense microstructure and less defects within the modified coating. However, large pores or defects are formed in coating with 0.8 wt.% YNPs, compromising the coating performance, as shown in Figure 3e,f. In the PTA process, due to the technical characteristics of rapid heating and cooling, large columnar crystals grow vertically from the bottom of the molten pool to the center of the coating, while the addition of YNPs will induce the dismember of dendritic crystals, so as to refine the grains and enhance the properties of the coating [23].

The XRD patterns of alloy powder and coating sample are shown in Figure 4. Figure 4a shows that the main phases of all alloy powders are $\alpha$-Fe phase (PDF#34-0529), Fe-Cr phase (PDF#85-1410), and Taenite phase (PDF#47-1417). The decrease in $\gamma$-Fe peak was due to the rapid cooling process during solidification [20,30]. Figure 4b shows that the main phases of all coatings are $\alpha$-Fe phase (PDF#34-0529), Fe-Cr phase (PDF#85-1410), and Cr phase (PDF#88-2323), corresponding to the diffraction peaks at 44.5°, 64.5°, and 43.5°, respectively. Notably, for Cr, it exhibits an obvious enrichment in the HSC0 coating samples [31]. However, the Cr phase in HSC02, HSC04, HSC06, and HSC08 samples decreased significantly with the addition of $Y_2O_3$. The addition of $Y_2O_3$ nanoparticles

provides sufficient undercooling [29] for PTA coating in the rapid solidification process, which promotes the diffusion rate of Cr and reduces non-equilibrium segregations [22,31]. Several weak XRD peaks are shown in HSC08 specimens, corresponding to the $Y_2O_3$ phase (PDF#65-3178) with a crystal plane of (222), (125), and (440), which peaked at 29.2°, 46.9, and 48.5°. It suggested that the extra $Y_2O_3$ nanoparticles will exist in the steel matrix to form inclusion, and its melting point is higher than that of the steel matrix.

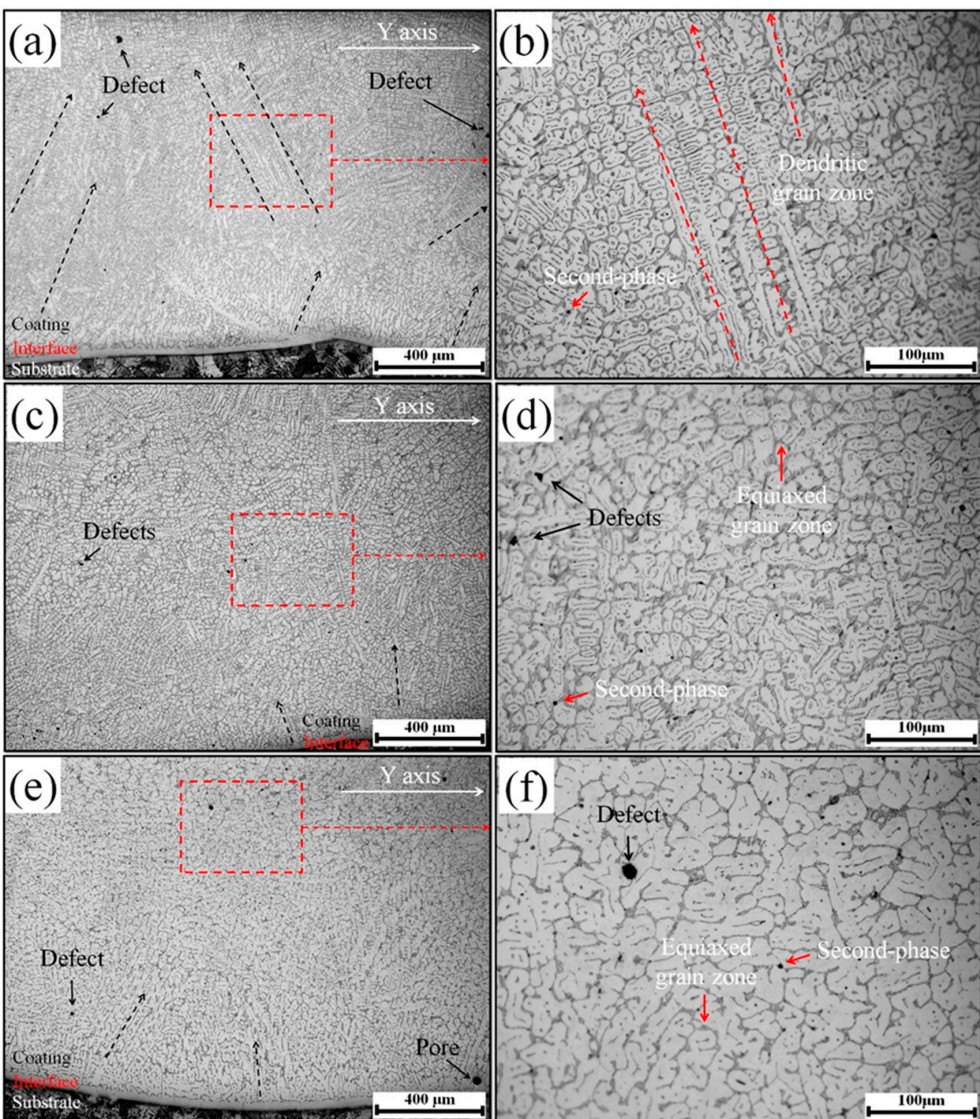

**Figure 3.** The cross-sectional morphologies of the coating samples: (**a**) HSC0, (**b**) Enlarged image of (**a**), (**c**) HSC04, (**d**) Enlarged image of (**c**), (**e**) HSC08, (**f**) Enlarged image of (**e**).

To further understand the influence of the Y element on 14CrSiMnV alloy coating, the EDS maps are measured to analyze the elemental composition of the secondary phase particles. Sphere-like secondary phase particles in HSC0 sample (Figure 5a) mainly consist of Si and O element. However, Figure 5b,c shows the secondary phase particles in the specimens modified with 0.4 wt.% and 0.8 wt.% YNPs, confirming the presence of Y, Mn, Si, and O. Hence, Y element segregated into Mn-oxides, followed by Y element playing a role in purifying grains and grain boundaries, which is consistent with XRD data. Furthermore, with the addition of YNPs, 18.9–23.0 wt.% of Y, 2.4–5.4 wt.% of Mn, and 18.2–24.5 wt.% of O, EDS was detected in the secondary phase particles, as shown in Table 3. This is due to the active chemical properties of the Y element, which has a strong affinity with O, Si, and Mn. The ICP mass spectrometer and OND determinate are measured to analyze the Y and

O content of the 14CrSiMnV alloy coating, and it was found that the Y and O contents in the coating increased with the increase of YNP content (Table 4).

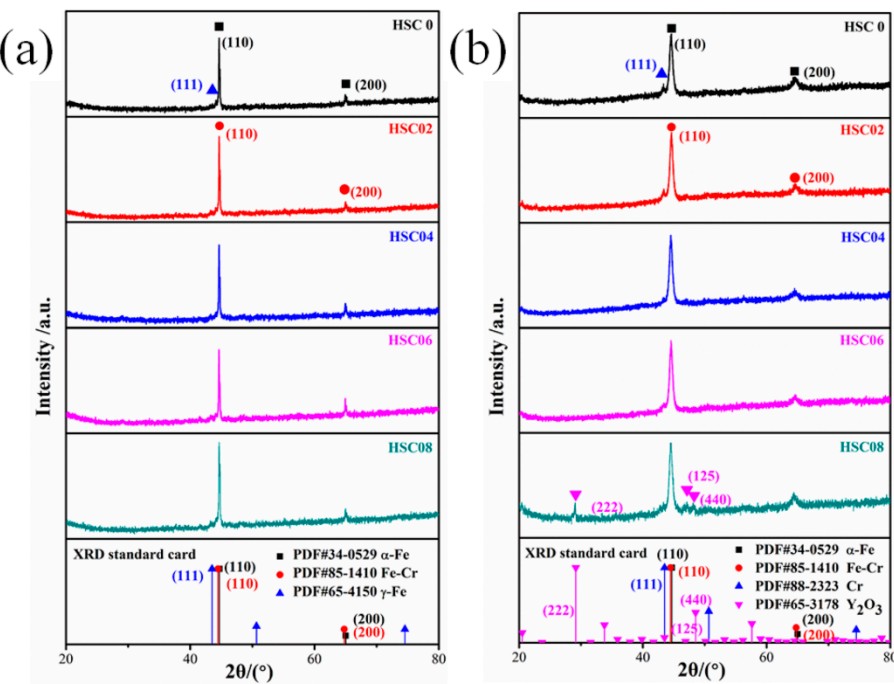

**Figure 4.** (**a**) XRD spectra and phase composition of the alloy powder samples, (**b**) XRD spectra and phase composition of the coating samples.

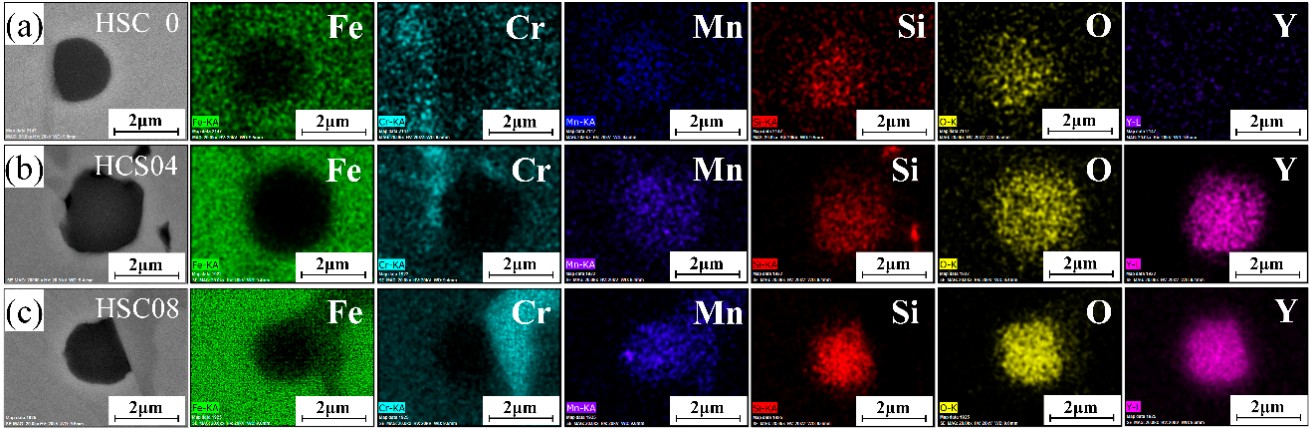

**Figure 5.** Backscattered electron image and the element composition of secondary phase in the coating samples: (**a**) HSC0, (**b**) HSC04, (**c**) HSC08.

**Table 3.** Normalization element of the secondary phase.

| EL | Fe/at.% | Cr/at.% | C/at.% | Al/at.% | Mn/at.% | Y/at.% | Si/at.% | O/at.% |
|---|---|---|---|---|---|---|---|---|
| HSC0 | 28.86 | 6.01 | 28.53 | 0.12 | 2.93 | - | 9.12 | 24.43 |
| HSC02 | 3.43 | 1.08 | 52.42 | 0.30 | 2.06 | 5.33 | 3.48 | 31.90 |
| HSC04 | 5.96 | 1.50 | 59.07 | 0.79 | 0.89 | 4.34 | 1.99 | 25.46 |
| HSC06 | 12.19 | 3.69 | 35.18 | 1.12 | 1.97 | 7.07 | 3.95 | 34.83 |
| HSC08 | 18.40 | 3.99 | 28.70 | 0.64 | 2.61 | 6.30 | 4.50 | 34.86 |

**Table 4.** O and Y element distribution of the coating samples.

| Sample | HSC0 | HSC02 | HSC04 | HSC06 | HSC08 |
|---|---|---|---|---|---|
| O(wt.%) | 0.016 | 0.018 | 0.020 | 0.026 | 0.026 |
| Y(wt.%) | 0 | 0.014 | 0.016 | 0.019 | 0.023 |

In the conventional $Y_2O_3$ addition mechanism, the compounds formed with dissolved Y ions as a slag float on the melt surface of PTA coating, playing the role of slag removal, tissue purification, porosity, and crack elimination [11,12,32]. On the other hand, abundant $Y_2O_3$ can provide conditions for the formation of Y oxides and impurity compounds to aggregate and grow up, leading to a decrease in the number of effective crystal nuclei, resulting in coarsening of the tissue and an increase in the number of inclusions. However, instead of the slag in modified coating, dissolved Y forms compound with Mn, Si, and O as secondary phase particles, while Y (0.180 nm), Mn (0.127 nm), and Si (0.146 nm) are suited to generate a substitutional solid solution during the rapid solidification process, which is relatively similar to the solid solution of Fe phase (0.172 nm), and provides abundant nucleation sites [33,34].

### 3.2. Mechanical Characterization

Figure 6 exhibits the tensile performance of HSC specimens. It is revealed that all test specimens display linear elastic strain curves. The tensile strength of the alloy coating specimen is 667 MPa (HSC0), 810 MPa (HSC02), 1281 MPa (HSC04), 1066 MPa (HSC06), and 946 MPa (HSC08), respectively. The elongation of the alloy coating specimens is 2.1% (HSC0), 2.8% (HSC02), 3.2% (HSC04), 3.0% (HSC06), and 2.9% (HSC08), respectively, as shown in Table 5. The tensile strength of the modified specimen (HSC04) increased by 92.0%, and the toughness of the coating increased by 55.6%. The YNP-modified alloy coating possesses higher tensile strength and elongation than the unmodified coating. One should note that grain refinement improves the tensile strength and ductility of the alloy coatings. During the PTA forming process, the Y element is combined with Mn, Si, and O elements to form new Y-Mn-Si-O compounds. These compounds are dispersed in the coating and become hetero-nucleation sites for grain nucleation, promoting grain nucleation and resulting in increased concentration equiaxed grains and short-term columnar crystals. The dispersion of the secondary phase particles in the coating results in a stress field, which hinders the deformation and enhances the mechanical properties of the modified coating.

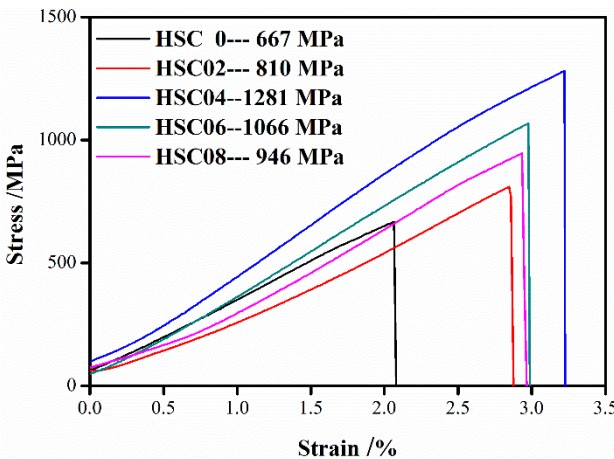

**Figure 6.** The stress–strain curve of the coating samples.

Figure 7 shows the fracture morphology of the tensile specimens. Figure 7a displays a long-range river-like pattern and fan-sliding morphology in the cross-section of HSC0 fracture sample. On the other hand, a mass of cleavage planes and step features, with

secondary micro-cracks and pores, are observed on the surface of HSC0 sample (Figure 7b), confirming the occurrence of cleavage and brittle fracture. It is revealed that the cleavage fracture occurred along the growth direction of long-range dendritic solidification microstructures, because cleavage and slip easily occur in microstructurally uneven regions of BCC-phase steel under loading, leading to linear elastic strain curves and cleavage fracture with low tensile properties [22].

**Table 5.** Tensile performance parameters of the coating samples.

| Samples | Modulus of Elasticity (Gpa) | Tensile Strength (Mpa) | Yield Strength (Mpa) | Elongation (%) |
|---------|------------------------------|-------------------------|----------------------|----------------|
| HSC0 | 292.5 | 667.3 | 113.3 | 2.1 |
| HSC02 | 263.8 | 810.0 | 82.7 | 2.8 |
| HSC04 | 367.3 | 1281.0 | 148.3 | 3.2 |
| HSC06 | 342.3 | 1066.9 | 99.1 | 3.0 |
| HSC08 | 297.5 | 946.6 | 108.8 | 2.9 |

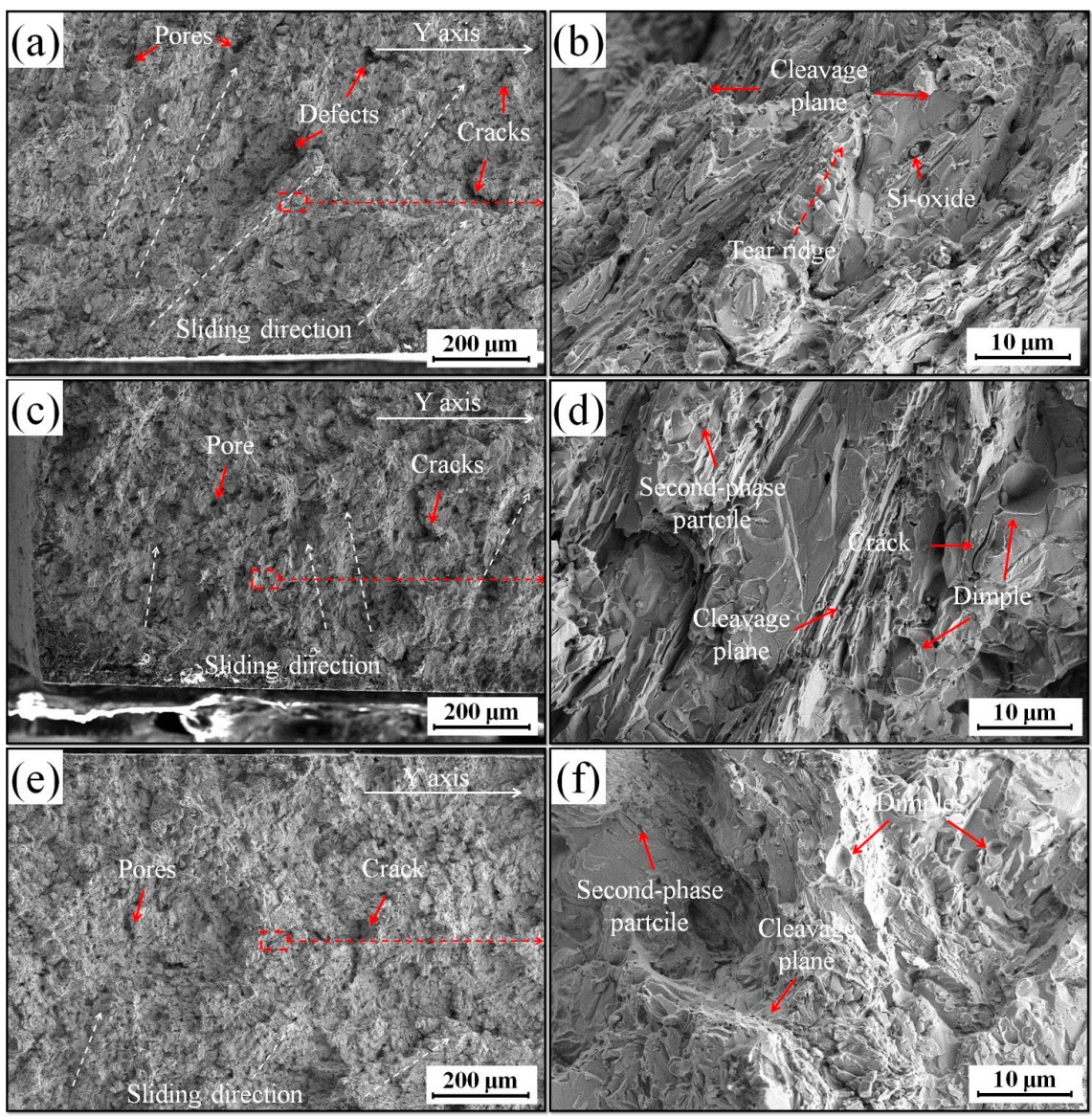

**Figure 7.** SEM morphology of fracture of the tensile specimen of the coating samples: (**a**) HSC0, (**b**) Enlarged image of (**a**), (**c**) HSC04, (**d**) Enlarged image of (**c**), (**e**) HSC08, (**f**) Enlarged image of (**e**).

Moreover, short-range river patterns, small cleavage steps, and dimples in the sliding direction were observed in the modified coating, as shown in the fracture morphology of HSC04 (Figure 7c,d) and HSC08 (Figure 7e,f) specimens, confirming the failure mechanism was based on hybrid-type with quasi-cleavage, brittle fracture, and granular ductile fracture [35]. On the one hand, extension sliding cracks were disassembled and blocked by short-range dendritic solidification and enlarged equiaxed grain zone. It is relatively easy to hinder the extension of cleavage steps during the tensile process due to the difference between secondary phase particles and coating in the elastic–plastic region and bonding ability, resulting in small cleavage steps under the influence of tensile stress. As a result, the brittle capacity of modified specimens is significantly improved. The addition of YNPs improves the internal grain structure of the coating, which plays a role in fine grain strengthening and second phase strengthening, while the dispersion distribution of the second phase particles hinders the expansion of slip dislocation and improves the strength of the coating [36]. However, excessive amounts of YNPs increase O content within the Fe phase (Table 4), resulting in large pores and micro-cracks (Figure 7d), and thereby decreasing the tensile strength of HSC08.

Figure 8 shows the longitudinal section morphology of the tensile specimens. The crystallographic facets with the size of a few microns at longitudinal section of the tensile specimens can be observed, as shown in Figure 8b–f. During the tensile process, the crack expands along the grain boundary and forms the crystallographic facets on the fracture surfaces [29]. Thus, the larger the grain size, the longer the crystallographic facets. The addition of YNPs will refine the grains, hinder the slip dislocation, and enhance the tensile property of the coating [23]. As a result, the crystallographic facets of HSC04 and HSC08 is smaller than that of HSC0.

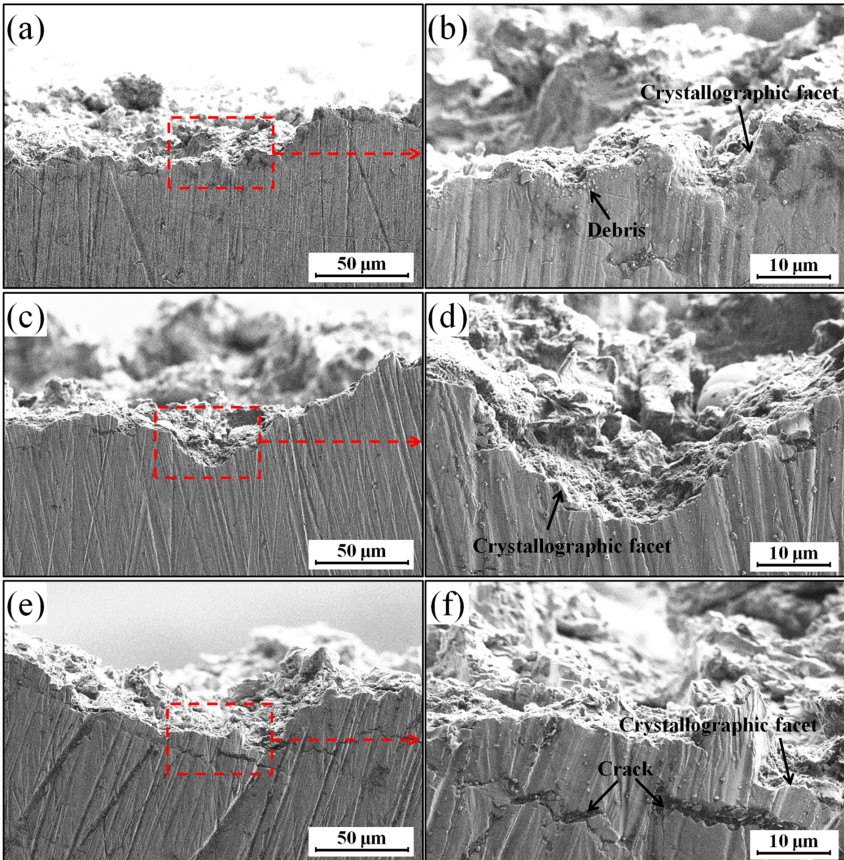

**Figure 8.** SEM morphology of longitudinal section of the tensile specimen of the coating samples: (**a**) HSC0, (**b**) Enlarged image of (**a**), (**c**) HSC04, (**d**) Enlarged image of (**c**), (**e**) HSC08, (**f**) Enlarged image of (**e**).

Figure 9 shows the microhardness along the cross-section of coatings. The average microhardness of the coating specimen is 627 HV0.1(HSC0), 658 HV0.1(HSC02), 698 HV0.1(HSC04), 669 HV0.1(HSC06), and 642 HV0.1(HSC08). Moreover, the microhardness of the modified coating increased with the YNPs content. When 0.4 wt.% YNPs are added, the modified coating exhibited the highest microhardness distribution, which is 11.3% higher than the unmodified coating. During the solidification process, the Y-Mn-Si-O compound disperses in the grains of coating and produces a pinning effect, inhibiting the grain growth and playing a role in grain strengthening and secondary phase strengthening. However, with the increase of YNPs content, excessive Y increases the viscosity of molten metal in the cladding pool and deteriorates the fluidity of molten metal, forming a large number of inclusions at grain boundaries, weakening the binding force between grains, thereby reducing the mechanical strength of the coating, including Vickers microhardness.

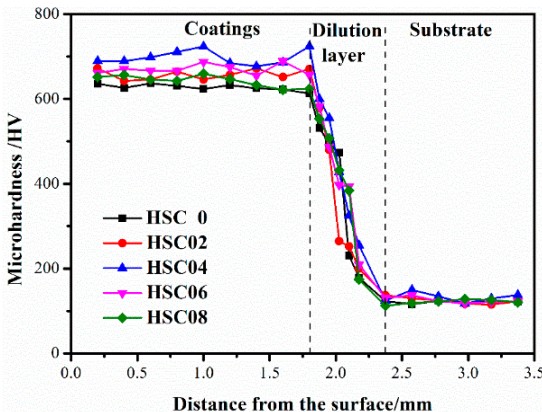

**Figure 9.** The microhardness distribution of the coating samples.

Figure 10 shows reciprocating sliding wear surface morphology of the coatings under a low-speed dry friction sliding. It displays a fish-scale pattern on the wear surface of all coatings, indicating the plastic deformation and adhesive wear characteristics due to delamination [37,38]. As shown in Figure 10a, both compact particles and deep grooves can be observed on uneven worn surfaces of the HSC0 specimen, whereas micro-plowing is the main feature of abrasive wear [37] due to third-body interactions. Owing to the dendritic solidification along the *z*-axis during the fast solidification and cooling process, a fine well-aligned structure is formed with a high-temperature gradient, leading to a large friction coefficient gap between the build direction (*z*-axis) and cladding direction (*x*-axis) of the PTA coating [3]. During the initial stage of sliding wear process, abundant debris is produced along the *x*-axis of coating due to brittle fracture and micro-cutting, promoting the adhesion and abrasive wear friction of unmodified coatings. On the other hand, HSC02 (Figure 10b), HSC04 (Figure 10c), and HSC06 (Figure 10d) exhibited abrasive wear, originating from the uniform microstructure and low amounts of debris. It is worth noting that the dendritic crystals inside the coating transform into equiaxed crystals after the addition of an optimal amount of YNPs, resulting in a lower friction coefficient and similar sliding wear resistance in different directions and less debris [39,40]. This is due to the addition of YNPs, improving the grain structure, hindering the growth of columnar grains, making the inner structure of the coating close, and improving the wear resistance of the coating. However, excessive YNPs will lead to superfluous O content and inclusion enrichment at grain boundaries in the coating, and the uneven internal composition will lead to reduce wear resistance. The EDS maps were measured to analyze the elemental composition of the reciprocating wear surface, as shown in Table 6. The results display that no new element (such as N) was introduced into the coating surface during the reciprocating friction process, while the wear resistance of the coating is improved by added small amount of YNPs.

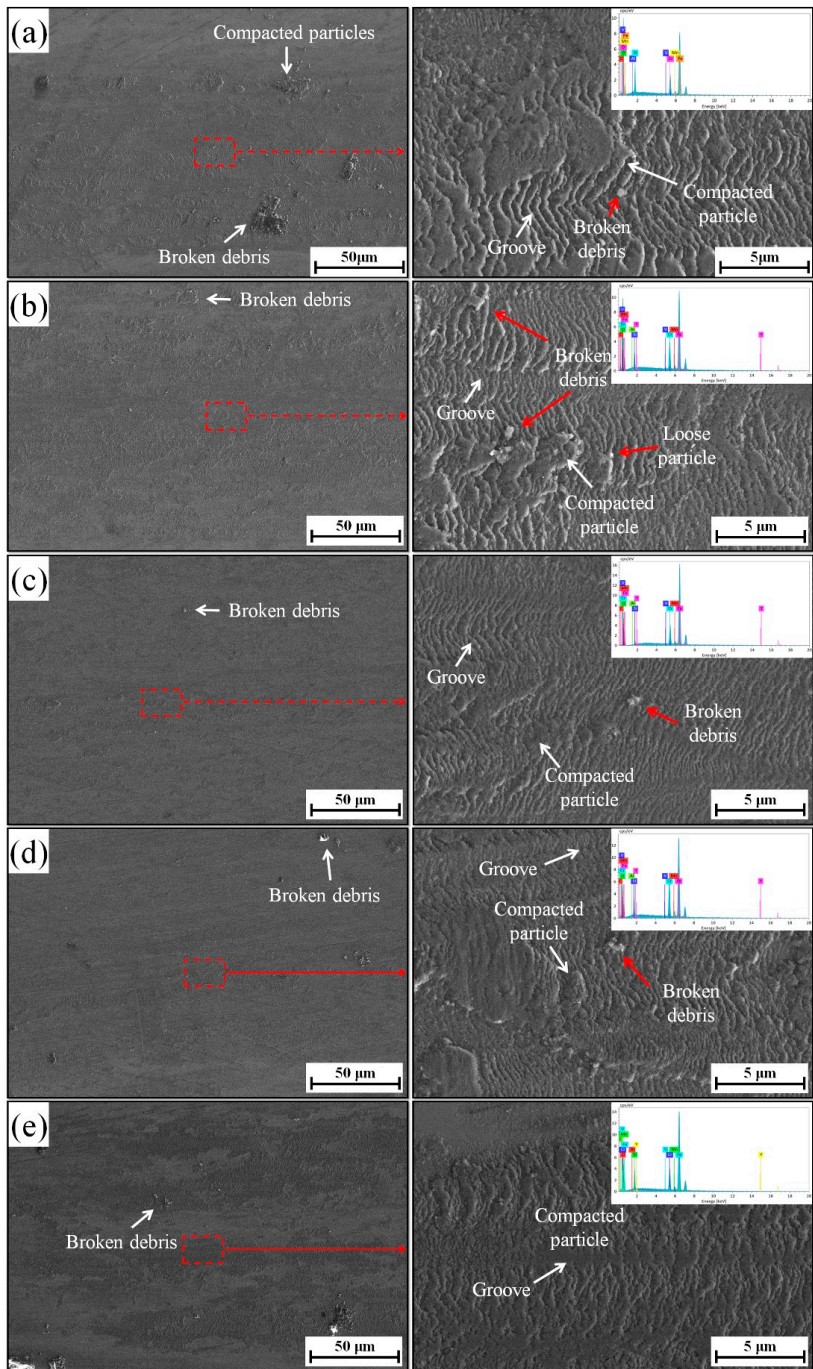

**Figure 10.** Wear surface image of the coating sample: (**a**) HSC0, (**b**) HSC02, (**c**) HSC04, (**d**) HSC06, (**e**) HSC08 (the inset image shows the corresponding EDS analysis).

**Table 6.** Normalization elemental composition of the reciprocating wear surface.

| EL | Fe/at.% | Cr/at.% | C/at.% | Al/at.% | V/at.% | Mn/at.% | Y/at.% | Si/at.% | O/at.% |
|---|---|---|---|---|---|---|---|---|---|
| HSC0 | 26.46 | 5.11 | 13.57 | 0.75 | 0.20 | 0.34 | - | 5.52 | 48.05 |
| HSC02 | 30.11 | 6.47 | 18.07 | 0.13 | 0.20 | 0.28 | 0.06 | 3.05 | 41.63 |
| HSC04 | 72.97 | 10.32 | 7.01 | 0.28 | 0.30 | 0.60 | 0.11 | 3.93 | 4.47 |
| HSC06 | 37.85 | 10.44 | 22.10 | 0.24 | 0.40 | 0.12 | 0.14 | 4.38 | 24.34 |
| HSC08 | 44.69 | 7.03 | 11.55 | 0.26 | 0.23 | 0.37 | 0.13 | 6.07 | 29.65 |

Figure 11 plots the weight loss–wear cycle curves and relative wear resistance coefficient of different coatings. During the initial stage of friction process, amounts of abrasive and debris particles are peeled off by brittle fracture from the coating surface, resulting in cracks or scars. Then, such particles are brought into the grooves to promote microcutting between surrounding matrix and eutectic structure during third-body interactions, resulting in a plastic deformation phenomenon and wear loss. The wear weight loss of specimens is found to be 0.1345 g (Q235), 0.0755 g (HSC0), 0.0752 g (HSC02), 0.0587 g (HSC04), 0.0693 g (HSC06), and 0.0732 (HSC08). The wear loss of HSC04 coating is the lowest, which is 56.3% and 22.2% less than the Q235 and unmodified coating, respectively.

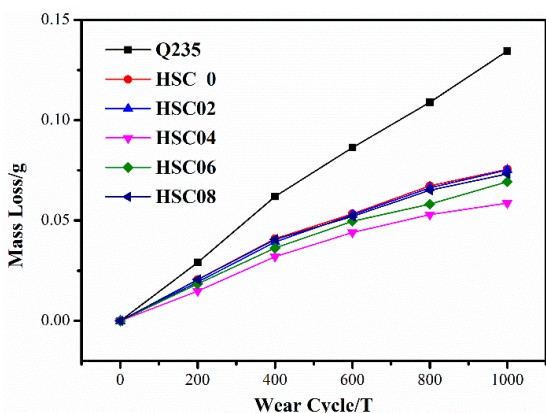

**Figure 11.** Weight loss–wear cycle curves.

Meanwhile, the relative wear resistance coefficient (RWSc) of Q235 steel was set at 1, while the RWSc of modified coating was found to be 1.7788 (HSC0), 1.7859 (HSC02), 2.2827 (HSC04), 1.9380 (HSC06), and 1.8347 (HSC08). The RWSc of the HSC04 specimen is increased by 28.3%. Figure 12a displays a worn surface with deep and coarse scratches, as well as large irregular spalling pits for unmodified coatings. In contrast, shallower friction scratches and smoother slender furrows were exhibited in the modified coating due to lesser amounts of brittle fracture and debris. It can be ascribed to a decrease of dendritic grains and enlarged equiaxed grains in the modified coatings, resulting in the transformation of the wear mechanism from plastic deformation to slight peeling.

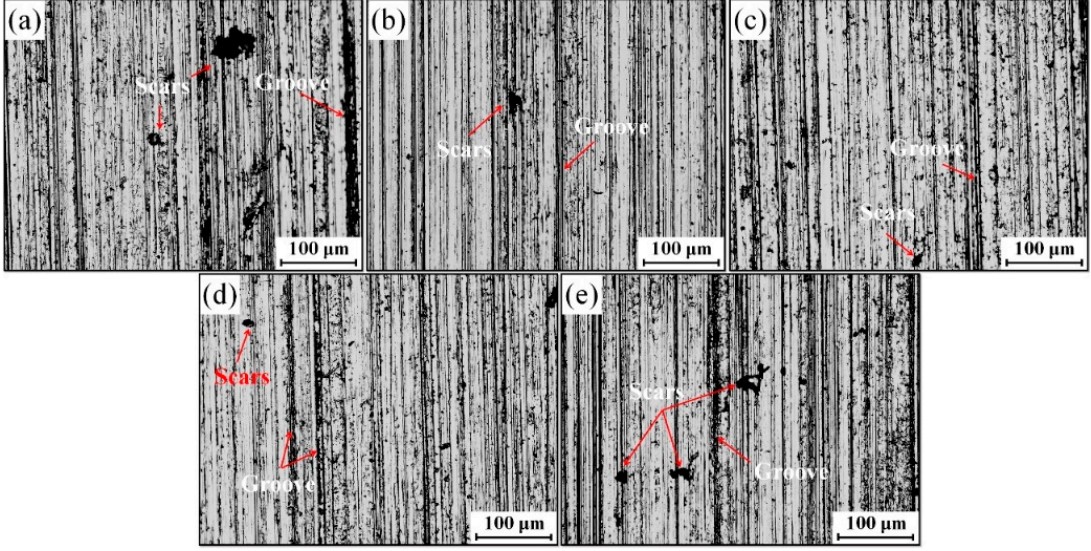

**Figure 12.** Worn surface image of the coating samples: (**a**) HSC0, (**b**) HSC02, (**c**) HSC04, (**d**) HSC06, (**e**) HSC08.

## 4. Conclusions

This work fabricated 14CrSiMnV steel coatings with small addition of YNP by PTA and explored the synergistic influence of YNPs addition and Y-Mn-oxide secondary phase on microstructure, tensile properties, and wear resistance. The results revealed that the presence of YNPs declined the dendritic solidification structure and enlarged the equiaxed grains. Moreover, part of Y combined with Mn, Si, and O elements to form Y-Mn-Si-O compound as a secondary phase reinforcer, resulting in grain strengthening and secondary phase strengthening, and leading to smaller cleavage plane, less third-body interactions and plastic deformation of the coating. As a result, the tensile properties, microhardness, and wear resistance of the modified steel coating have been obviously improved. The optimal amount of YNPs (0.4 wt.%) resulted in an increase of 92.0% in tensile strength, an increase of 55.6% in elongation, an increase of 11.3% in microhardness, a reduction of 22.2% in wear weight loss, and an increase of 28.3% in relative wear resistance. However, excessive YNPs will introduced superfluous O content, oxide slag, and defect in the coating, resulting in decline of the mechanical and wear properties. Overall, the proposed strategy of combining Y, Mn, Si, and O elements to form secondary phase compounds effectively solved the problems of nonequilibrium segregations at dendritic boundaries (in the interaxial space) and properties degradation.

**Author Contributions:** Conceptualization, J.Y. and Y.S.; methodology, J.Y.; software, J.Y.; validation, Y.S. and L.Y.; formal analysis, X.L.; investigation, X.L.; data curation, L.Y.; writing—original draft preparation, J.Y.; writing—review and editing, X.S.; visualization, Y.S.; supervision, F.L.; funding acquisition, W.C. All authors have read and agreed to the published version of the manuscript.

**Funding:** This research was funded by the National Nature Science Foundation of China, grant number 51964035 and the Natural Science Foundation of Inner Mongolia Autonomous Region, grant number 2020LH05017.

**Institutional Review Board Statement:** Not applicable.

**Informed Consent Statement:** Not applicable.

**Data Availability Statement:** Not applicable.

**Acknowledgments:** This research was funded by the National Nature Science Foundation of China, grant number 51964035 and the Natural Science Foundation of Inner Mongolia Autonomous Region, grant number 2020LH05017.

**Conflicts of Interest:** The funders had no role in the design of the study; in the collection, analyses, or interpretation of data; in the writing of the manuscript, or in the decision to publish the results.

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
