# Peer review of "Influence of Y Nano-Oxide and Its Secondary Phase on Microstructure, Mechanical Properties, and Wear Behavior of the Stainless Steel Coatings Fabricated by Plasma Transfer Arc"

_metals, doi:10.3390/met12060942_

Round 1

Reviewer 1 Report

In the present research, the authors try to investigate the precipitates and mechanical properties of the plasma-cladded steel coating. The paper exhibits some results but there are some questions.

  1. The authors titled the paper as “Enhance mechanical properties of plasma-cladded steel coatingby Y and Mn based secondary phase oxide”. Based on the title, the secondary oxide particles should be the focus. However, the characterization on the secondary phase oxides is so less. Actually, the Y oxides is added as the original powder. Then its evolution would be important for the mechanical properties of the steel coating. Their distribution and structure could influence the microstructure and mechanical properties of the as-fabricated materials. The authors could refer the previous research “Microstructure evolution and mechanical properties of Ni3Al/Al2O3 composite during self-propagation high-temperature synthesis and hot extrusion. Materials Science and Engineering: A 2012,555: 131-138”. Additionally, the wear properties could be added in the title.
  2. In the experimental, the chemical composition of the Q235 steel should be given for better understanding of the content.
  3. In the experimental or content, the authors are suggested to provide the XRD analysis on the steel powders with different Y2O3 oxides content for understanding microstructure of the cladding layer.
  4. In the content, the authors provide the SEM observations on the cladded coating, but few on the interface of coating and matrix. It wonders what the interaction between the coating and the matrix? In fact, the provided microstructure indicates the influence of the initial powder content and matrix. Moreover, the phase constituent and secondary phase distribution should be analyzed in detail. The authors could take more analysis and refer the recent research “Improving surface quality and superficial microstructure of LDED Inconel 718 superalloy processed by hybrid laser polishing. Journal of Materials Processing Technology 2022,300:117428”.
  5. In the Figure 3, what is the last XRD pattern? There should be some explanation in the figure caption or some description in the figure.
  6. In the Figure 5, the x-axis should be moved to the right more or less. Then the yield stage of the tensile curve would be more clear.
  7. In the content, the authors provide the surface fracture of the tensile specimen. Though some features could be observed in these observations, however it could not reflect the influence of the secondary phase or oxides. The authors are suggested to observe the cross-sectional microstructure of the failed specimen adjacent to the fracture. The authors could refer the previous research “Microstructural characteristics and mechanical properties of the hot extruded Mg-Zn-Y-Nd alloys.Journal of Materials Science & Technology 2021,60: 44-55”.  
  8. In the content, the authors have provide the worn surface of the coating with different Y2O3 oxidesaddition in Figure 8. Though some feature could be observed, however what is the influence of the Y2O3 oxides? The authors are suggested to perform the elemental distribution analysis and cross-sectional microstructure of the wear tested specimens. The authors could refer the previous researches “Influence of layer number on microstructure, mechanical properties and wear behavior of the TiN/Ti multilayer coatings fabricated by high-power magnetron sputtering deposition. Journal of Manufacturing Processes 2021,70:529-542” and “Investigation on microstructure and wear behavior of the NiAl-TiC-Al2O3 composite fabricated by self-propagation high-temperature synthesis with extrusion. Journal of alloys and compounds 2013,554: 182-188”.
  9. In the Figure 9, what is the difference between the two figures. The data of Figure 9b could be found in the Figure 9a. The authors could delete the Figure 9b.
  10. The authors give the worn surface of the coating specimens. It wonders what is the difference between Figure 10 and Figure 8? The authors could give more detailed observations on the worn surface.
  11. In the introduction, the description could be improved. Based on the content, the authors mainly focus on the surface modification. Actually, there are many kinds of surface modification method to improve the properties of the substrate materials. The authors could refer the previous researches “Microstructure and mechanical properties of the Ag/316L composite plate fabricated by explosive welding. Journal of Manufacturing Processes 2021,64: 265-275”.  
  12. Some discussion on the results could be performed to reveal the influence of the Y2O3 oxides addition. In addition, the relationship between the microstructure and mechanical/wear properties could be analyzed in the content.

Reviewer 2 Report

The work is quite interesting and of importance for the development of steel´ covering materials, the aim is clearly stated, the presented methodology is adequated, and it is necessary to better describe the tensile test specimen size end geometry. From the results, it was possible to identify and describe the microstructural strengthening micromechanisms.

Please find the suggestion in the attached file.

Reviewer 3 Report

The article is a detailed ionvestigations of a complex multi-phase rapidly quenched  cast coating. Dispersed particles of yttrium oxide were introduced into the coating, which partially solved into a solid solution. This allowed yttrium to perform several functions as a solid solution hardener, surfactant and precipitation hardening agent. All obtained results are scientifically substantiated. Some conclusions require minor clarifications. According to its relevance and content, the article can be published.

  1. Lines 65-68 in section 1 should be deleted, "The results reported here are for an optimal addition of YNPs (0.4 wt. %), the mechanical properties of the steel coating are improved, as indicated by the increase of 92.0% in the tension strength, increase of 55.6% in the elongation, increase of 11.3% in the microhardness, decrease of 22.2% in wear weight loss, and increase of 28.3% in relative  wear resistance.", especially since they repeated in the abstract and conclusion. The phrase should be changed. It is advisable to write that the purpose of the work is to find the optimal alloying of the coating to improve its performance properties.
  2. A more serious argument is required for the presence of Si, Y, Mn in the solid solution and in secondary phases. We kindly request the authors to provide data of the chemical composition of secondary phases in atomic percents (Tables 2 and 3). This would allow them to more accurately predict the stoichiometry of oxides. In this case, the oxygen content will allow Si, Mn, Y to be bound into oxides, and their the excess is attributed to the solid solution.. Reasoning about the atomic radii of the elements is not entirely correct. Unpolarized ionic radii must be taken into account. Since the chemical elements in the crystal lattice are present in the form of ions, not free atoms.
  3. The balance of each row in table 2 is not 100%. In this case, the authors claim a high accuracy of spectral analysis of 1/100%. These results need to be refined. Please explain how the data on O and Y content in the coating were obtained, which are presented in Table 3. The method and area size for analysis should be presented in section 2.3.
  4. It seems to me that the expression "effectively solved the problems of elemental segregation at the grain boundary and properties degradation" does not quite correctly. The article investigate a cast structure, for which one can speak of non-equilibrium segregations at dendritic boundaries (in the interaxial space) in contrast to grain boundaries equilibrium segregations .

Round 2

Reviewer 1 Report

In the present state, the authors have revised the paper and answered the reviewer's questions. Now, the paper is improved and could be accepted.